# Genetic Blueprints in Lung Cancer: Foundations for Targeted Therapies

**DOI:** 10.3390/cancers16234048

**Published:** 2024-12-02

**Authors:** Andra Dan, Livia-Malina Burtavel, Madalin-Codrut Coman, Ina-Ofelia Focsa, Simona Duta-Ion, Ioana-Ruxandra Juganaru, Andra-Giorgiana Zaruha, Patricia-Christina Codreanu, Irina-Maria Strugari, Iulian-Andrei Hotinceanu, Laurentiu-Camil Bohiltea, Viorica-Elena Radoi

**Affiliations:** 1Department of Medical Genetics, “Carol Davila” University of Medicine and Pharmacy, 020021 Bucharest, Romania; andra.dan@rez.umfcd.ro (A.D.); livia-malina.burtavel@rez.umfcd.ro (L.-M.B.); ina.focsa@umfcd.ro (I.-O.F.); simona-gabriela.duta@rez.umfcd.ro (S.D.-I.); ioana-ruxandra.calapod@rez.umfcd.ro (I.-R.J.); andra-giorgiana.zaruha@rez.umfcd.ro (A.-G.Z.); patricia-christina.codreanu@rez.umfcd.ro (P.-C.C.); maria-irina.strugari@rez.umfcd.ro (I.-M.S.); iulian.hotinceanu@s.unibuc.ro (I.-A.H.); laurentiu.bohiltea@umfcd.ro (L.-C.B.); viorica.radoi@umfcd.ro (V.-E.R.); 2“Alessandrescu-Rusescu” National Institute for Maternal and Child Health, 20382 Bucharest, Romania

**Keywords:** lung cancer, NSCLC, SCLC, genetics, pathways, alterations, biomarkers, diagnosis, therapies, personalized medicine

## Abstract

This article explores the molecular pathways involved in lung cancer, alongside genomic alterations and genetic syndromes associated with the disease. It also reviews testing methods and their role in advancing personalized treatment approaches, offering insights into current therapeutic strategies.

## 1. Introduction

Lung cancer, a malignant neoplasm originating from the epithelial cells of the lung, is characterized by its aggressive growth and poor prognosis, making it a leading cause of cancer-related mortality globally [1,2]. In the European Union (EU), lung cancer represents a significant public health challenge, accounting for approximately one-fifth of all cancer-related deaths. In 2020, lung cancer was responsible for the deaths of nearly a quarter of a million individuals within the EU, constituting 4.5% of the overall mortality rate and 19.8% of all cancer-related deaths. The epidemiology of lung cancer exhibits a gender disparity, with males demonstrating a higher mortality rate at 5.9%, nearly double that of females at 3.0% [3,4]. 

The substantial mortality associated with lung cancer has driven intensive research into novel treatment options that leverage the genetic basis of the disease. Advances in genomic research have illuminated the molecular pathways driving lung cancer, leading to the development of targeted therapies [5]. Targeted therapies are designed to specifically interact with molecular alterations—such as mutations in EGFR and ALK genes—that promote cancer cell growth and survival. Unlike traditional chemotherapy, which affects both healthy and cancerous cells, targeted therapies selectively inhibit pathways essential to tumor progression, resulting in enhanced efficacy and reduced toxicity [6].

For lung cancers, targeted therapies, including tyrosine kinase inhibitors (TKIs), have shown particular effectiveness by blocking the signaling functions of mutated genes critical to tumor growth [7]. Integrating biomarkers into clinical practice further enables a personalized approach, where treatment selection is informed by the genetic profile of the patient’s tumor, thereby improving survival rates and quality of life. This review provides an in-depth exploration of lung cancer from a genetic perspective, with a focus on the molecular pathways that drive its pathogenesis, highlighting advances that pave the way for precision therapy [7,8].

## 2. Classification of Lung Cancers

From a histological perspective, lung cancer is primarily divided into two main subtypes: small-cell lung cancer (SCLC) and non-small-cell lung carcinoma (NSCLC). Additionally, there are less common forms of lung cancer, such as carcinoid tumors, bronchial gland carcinomas, and sarcomatoid carcinomas (Figure 1). NSCLC constitutes approximately 85% of all lung cancer cases, while SCLC represents 10–15% of cases [9].

In 2021, the World Health Organization (WHO) revised the classification of lung tumors based on morphological characteristics, molecular abnormalities, and immunohistochemical profiles. The main categories of lung cancer classification now include epithelial tumors, lung neuroendocrine neoplasms, tumors of ectopic tissues, mesenchymal tumors specific to the lungs, and hematolymphoid tumors [10].

## 3. Genetic Predisposition and Associated Genes

Smoking is the principal cause of lung cancer, accounting for the majority of cases, though other factors such as air pollution, ionizing radiation, chronic pulmonary conditions, and occupational exposures also contribute, either independently or synergistically [11,12]. While cigarette smoking remains the leading cause of lung cancer, advances in cigarette design have introduced complexities in understanding its impact on cancer risk. Modern cigarettes often incorporate cellulose acetate filters, ventilation, and expanded tobacco to lower tar levels per cigarette, resulting in so-called “low-tar” options [13,14]. However, these modifications may affect cancer risk differently depending on the histological subtype of lung tumors, with studies suggesting distinct patterns between filter and non-filter cigarettes. Beyond conventional cigarettes, e-cigarettes and vaping products have emerged as potential contributors to lung cancer risk [15]. Research indicates that vaping aerosols contain known carcinogens, such as polycyclic aromatic hydrocarbons, heavy metals, and aldehyde-based flavoring agents, which may promote DNA damage. Although vaping is generally considered less harmful than traditional smoking, its long-term cancer risk remains uncertain and is the subject of ongoing research, especially as usage rises among younger populations and non-smokers [16,17].

Additionally, a significant number of lung tumors often occur in non-smokers, without identifiable external risk factors. Individuals with a family history of lung cancer, especially those with early-onset disease, exhibit an elevated risk. This stipulates that genetic factors play a crucial role in the development of lung cancer. 

Lung tumor cases associated with inherited genetic variations typically exhibit an autosomal dominant inheritance pattern. Most lung cancer cases are not linked to inherited genetic mutations, but rather arise from somatic mutations specifically within lung cells. However, there are several genetic syndromes known to present a risk for developing lung tumors (Figure 2). These genetic conditions vary in their level of association with lung cancer, reflecting a range of potential risks [18,19,20]. 

Cancer cells emerge from non-malignant tissues through the sequential accumulation of molecular alterations that drive proliferation, bypass growth suppression and apoptosis, and enable processes such as angiogenesis, invasion, and metastasis. Typically, cancer-related genes are categorized as either oncogenes, which promote tumorigenesis through gain-of-function mutations, or tumor suppressor genes (TSGs), which require loss-of-function mutations to lose their protective roles against cancer [21]. 

In lung adenocarcinoma, key oncogenes such as KRAS, EGFR, ALK, ERBB2, and BRAF are frequently mutated and are found in over 50% of cases [22,23]. Alongside these alterations, tumor suppressor genes like TP53, STK11, and CDKN2A are significantly impacted, with other suppressors such as NF1, ATM, APC, and RB1 also playing crucial roles in the development and progression of lung adenocarcinoma [23,24]. However, some genes can exhibit both oncogenic and tumor-suppressive functions depending on the cellular context, due to the expression of multiple isoforms and post-translational modifications [24]. This dual functionality underscores the complexity of cancer biology, where single genetic events can shift the balance from tumor suppression to oncogenesis.

In addition to the roles of oncogenes and TSGs in cancer development, another pivotal mechanism is the emergence of gene fusions, known as oncofusions (Figure 3). These hybrid genes result from genomic rearrangements such as deletions, inversions, translocations, or duplications, where two previously independent genes are juxtaposed [25,26]. 

Oncofusions arise primarily from these rearrangements and contribute to tumorigenesis by either deregulating existing cellular pathways or creating novel, oncogenic activities. With over 10,000 gene fusions identified in various human cancers, they represent a significant category of driver alterations in cancer biology [24,25,26]. The most common targetable fusions involve the anaplastic lymphoma kinase (ALK) gene, followed by those involving ROS1 and RET proto-oncogenes. These fusions provide critical targets for precision therapies in NSCLC patients [27].

## 4. Molecular Pathways and Therapeutic Strategies

Lung cancer is a complex and multifactorial disease, driven by intricate molecular mechanisms that regulate cell proliferation, survival, and metastasis. This complexity arises from numerous interconnected signaling pathways, each playing a distinct role in driving cancer progression and shaping the tumor microenvironment. Among the most critical pathways implicated in its pathogenesis are those governing cellular responses to growth factors, energy stress, and immune evasion [28,29]. These signaling cascades, including MAPK, JAK-STAT, and EGFR/PI3K/Akt/mTOR, orchestrate key oncogenic processes, while dysregulation in pathways like Wnt/β-catenin, VEGF/VEGFR, and NF-κB promotes tumor progression and angiogenesis. 

Moreover, aberrations in hypoxia signaling, apoptosis regulation, and immune checkpoints further enable cancer cells to adapt and evade therapeutic interventions, making these pathways essential targets in lung cancer treatment and research. By thoroughly examining how these disrupted pathways interact with one another, we can gain a deeper understanding of the complex mechanisms driving tumor biology, which not only enhances our knowledge of cancer progression but also uncovers new and potentially more effective therapeutic targets that could be exploited for treatment interventions.

### 4.1. The MAPK Pathway

The MAPK pathway, also known as the RAS-RAF-MEK-ERK pathway, is pivotal in transmitting external signals into cellular actions, regulating processes such as proliferation, survival, and metastasis. The pathway’s four primary cascades—ERK1/2, JNK, p38, and ERK5—each play specialized roles in response to stimuli like growth factors or environmental stresses. Active at all stages of tumor development, this pathway contributes significantly to cancer progression, especially through mutations like KRAS, which are found in 30% of lung adenocarcinomas [28,29,30]. Mutations in upstream components such as KRAS and BRAF lead to constitutive activation of ERK1/2, promoting uncontrolled cell growth, evasion of apoptosis, and resistance to therapy. Activated ERKs translocate to the nucleus, where they modulate transcription factors, altering gene expression to favor oncogenic processes (Figure 4). Notably, ERK signaling is associated with the upregulation of epidermal growth factor receptor (EGFR) ligands, such as TGF-α, establishing an autocrine feedback loop that sustains tumor progression [31].

The dysregulation of the RAS-RAF-MEK-ERK pathway contributes to the aggressive nature of these cancers, making it a key target for therapeutic strategies. The interaction between the RAS and TGF-β signaling pathways drives epithelial‒mesenchymal transition (EMT) and extracellular matrix (ECM) remodeling, which are key processes in cancer invasion and metastasis. The RREB1 gene, activated by RAS, primes EMT and fibrogenic gene enhancers, which are then activated by the TGF-β/SMAD pathway. Blocking RREB1 disrupts this process, preventing metastasis and presenting RREB1 as a potential target for cancer therapy [32,33]. MEK inhibitors, such as trametinib and selumetinib, have shown limited efficacy as monotherapies, but their combinations with other targeted therapies, including BRAF inhibitors (e.g., dabrafenib), have demonstrated improved outcomes in BRAF V600E-mutant NSCLC. These combination strategies enhance tumor response, prolong progression-free survival, and reduce resistance mechanisms, particularly in ALK-rearranged cancers. Further, preclinical studies indicate that dual MEK and ALK inhibition effectively suppresses tumor growth, highlighting the potential for combination therapies to overcome pathway-specific resistances [31,32,33,34].

### 4.2. The JAK-STAT Pathway 

The JAK-STAT pathway is a highly conserved signaling mechanism that is crucial for transmitting extracellular signals into the nucleus, ultimately leading to changes in gene expression (Figure 5). This pathway involves four JAK proteins (JAK1, JAK2, JAK3, and TYK2) and seven STAT proteins (STAT1, STAT2, STAT3, STAT4, STAT5A, STAT5B, and STAT6). The pathway is primarily activated by cytokines binding to their respective cell-surface receptors. This binding event triggers the activation of receptor-associated Janus kinases (JAKs). Once activated, JAKs phosphorylate specific tyrosine residues on the receptor, facilitating the recruitment of signal transducers and activators of transcription (STATs). In addition to cytokines, growth factors such as EGF can also activate STATs, expanding the pathway’s role in cellular signaling. Phosphorylated STATs then dimerize, typically through interactions between their SH2 domains and phosphorylated tyrosine residues [35,36]. These dimers translocate to the nucleus, where they bind to DNA and modulate the transcription of target genes. In the context of NSCLC, the JAK-STAT pathway, particularly STAT3 and STAT5, is frequently aberrantly activated. Persistent activation of STAT3 has been linked to enhanced tumor proliferation, survival, and metastasis. This constitutive activation supports oncogenic processes such as resistance to apoptosis, angiogenesis, and increased invasive potential, contributing to the aggressive nature of NSCLC [37,38].

Building upon this understanding, therapeutic targeting of the JAK/STAT pathway has emerged as a promising strategy in solid tumors, including lung cancer. Preclinical and clinical studies have demonstrated that JAK inhibitors, such as ruxolitinib, momelotinib, and AZD1480, exhibit potential efficacy as both monotherapy and in combination with other treatments like tyrosine kinase inhibitors (TKIs). In non-small-cell lung cancer (NSCLC), inhibition of JAK2 has been shown to reduce STAT3 phosphorylation, a critical step in preventing its nuclear translocation and subsequent transcriptional activation of oncogenic genes. Moreover, JAK inhibition has been proposed to counteract drug resistance mechanisms; for instance, targeting JAK2 in EGFR-mutant lung adenocarcinoma can restore the effectiveness of EGFR inhibitors like erlotinib and gefitinib by modulating interactions with SOCS5. These findings highlight the pathway’s therapeutic potential, not only in limiting tumor progression but also in enhancing the efficacy of existing cancer therapies [37,38].

### 4.3. The LKB1/AMPK/mTOR Pathway 

The LKB1/AMPK/mTOR pathway is a critical signaling cascade that regulates energy homeostasis, cell growth, and survival. Liver kinase B1 (LKB1) is a key regulator of this pathway, activating AMPK in response to cellular energy changes and acting as a metabolic checkpoint. This pathway also helps maintain redox balance, particularly through NADPH regulation. AMPK inhibits the mTOR pathway, which controls cell growth and metabolism, conserving energy during metabolic stress (Figure 6). Additionally, the pathway influences cell polarity, the cytoskeleton, and the cell cycle and functions as a tumor suppressor [39]. In NSCLC, the LKB1/AMPK/mTOR pathway plays a dual role. While LKB1 acts as a tumor suppressor by regulating cell polarity, energy metabolism, and inhibiting cell proliferation, its dysregulation is associated with tumor progression and metastasis. Loss of LKB1 function disrupts these regulatory mechanisms, leading to increased invasive and migratory capabilities of cancer cells.

Notably, while LKB1/AMPK signaling typically acts as a barrier to tumorigenesis by maintaining energy balance and inhibiting mTOR activity, its dysregulation can also confer survival advantages on cancer cells, particularly in the nutrient-deprived tumor microenvironment. Furthermore, genetic variations within this pathway, such as mutations in AKT and TSC1, have been associated with differential responses to chemotherapy in NSCLC patients, suggesting that the LKB1/AMPK/mTOR axis may also impact treatment outcomes [39,40]. Therapeutic targeting of the LKB1-AMPK pathway offers significant potential in NSCLC by suppressing tumor growth and enhancing treatment responses. AMPK agonists like metformin and AICAR activate this pathway, disrupting mTOR signaling and reducing cancer cell migration and proliferation. Preclinical studies have demonstrated that activating LKB1-AMPK signaling can delay tumor onset and inhibit metastasis, particularly in models with hyperactive mTOR. Emerging strategies, such as tankyrase inhibitors, further support the pathway’s role as a therapeutic target, highlighting its promise in combating tumor progression and drug resistance in lung cancer [41].

### 4.4. The EGFR/PI3K/Akt/mTOR Pathway

The EGFR/PI3K/Akt/mTOR pathway is integral to the progression of various cancers, including NSCLC. This pathway controls vital cellular processes such as metabolism, migration, cell survival, and proliferation, and its impact extends to modifying the tumor microenvironment [42]. Specifically, it facilitates the recruitment of inflammatory cells and stimulates angiogenesis, which is essential for tumor growth and metastasis (Figure 7). In lung cancer, mutations and amplifications in the PIK3CA gene are frequently observed, particularly in squamous cell carcinoma, and significantly contribute to tumorigenesis and disease progression. 

Moreover, the activation of Akt and mTOR, driven by mutations in PIK3CA, EGFR, or KRAS, is frequently associated with resistance to conventional therapies, further complicating treatment strategies. The EGFR/PI3K/Akt/mTOR pathway governs several processes in lung cancer, including genomic stability, cell survival, senescence, angiogenesis, proliferation, metastasis, and cellular metabolism. Therefore, any dysregulation or mutation within this pathway can substantially contribute to lung cancer progression. In NSCLC, one of the primary mutations occurs in receptor tyrosine kinases (RTKs), which act as the initial trigger for the pathway. Similarly, KRAS mutations can activate this pathway through alternate mechanisms. The pathway’s importance is further highlighted by its role in tumors with activating mutations, particularly those involving EGFR. Research shows that the Akt/mTOR signaling cascade remains constitutively active in approximately 67% of patients with EGFR mutations [42,43,44].

For patients with EGFR mutations, several targeted therapies have been developed, offering a range of benefits based on the specific mutation and patterns of resistance observed. One notable example is osimertinib, a third-generation EGFR tyrosine kinase inhibitor (TKI), which has shown significant efficacy, especially in patients with T790M-positive mutations. Due to its favorable progression-free survival (PFS) and overall survival (OS) outcomes, osimertinib is often considered a preferred first-line treatment for such patients [45,46,47].

### 4.5. The Keap1-Nrf2 Pathway 

The Keap1-Nrf2 pathway plays an important role in cellular defense by regulating the expression of detoxifying and antioxidant enzymes. Nrf2 (Nuclear factor erythroid 2-related factor 2) is a transcription factor that drives the expression of these protective genes, while Keap1 (Kelch-like ECH-associated protein 1) serves as its primary regulator. Under normal conditions, Keap1 binds to Nrf2, promoting its ubiquitination and degradation, thereby maintaining low Nrf2 activity [48]. However, under oxidative stress, modifications in Keap1 prevent this degradation, allowing Nrf2 to accumulate and activate the transcription of target genes involved in cellular protection (Figure 8).

In lung cancer, alterations in the Keap1-Nrf2 pathway, such as mutations or loss of Keap1 function, are common and significantly contribute to tumor development. These mutations often disrupt the Keap1-Nrf2 interaction, leading to increased Nrf2 activity and the upregulation of genes that promote tumor survival and progression [49]. Studies have shown that Keap1 mutations are frequent in lung squamous cell carcinoma, leading to persistent Nrf2 activation, which promotes drug resistance and tumor growth [48,49]. Inhibitors like brusatol have shown potential in reducing Nrf2 activity, thereby suppressing migration and invasion of NSCLC cells by modulating the RhoA-ROCK1 pathway. Furthermore, mutations in KEAP1 or NFE2L2 are associated with shorter survival in NSCLC patients, emphasizing the need for targeted therapies. Developing strategies to regulate Nrf2 activity without triggering adverse effects could offer significant clinical benefits [50].

### 4.6. The Wnt/β-Catenin Pathway 

The Wnt/β-catenin pathway is triggered when Wnt ligands bind to cell receptors, stabilizing β-catenin and allowing it to accumulate in the nucleus, where it promotes gene expression related to cell growth and survival. Without Wnt signaling, β-catenin is degraded (Figure 9) [51]. 

In lung adenocarcinoma cells, WIF1 overexpression has been shown to inhibit migration and invasion by downregulating Wnt/β-catenin signaling. Similarly, the secreted frizzled-related protein (sFRP) family antagonizes Wnt signaling by competing with Fzd receptors for Wnt ligands, while Dickkopf (DKK) family members like DKK-1 are often overexpressed in lung cancer tissues and associated with tumor progression and metastasis [52,53]. Additionally, the DKK-3 gene, when reactivated, can induce cell cycle arrest and apoptosis and improve cisplatin sensitivity in resistant cell lines. Oncogenes like TIPE and high mobility group protein 3 (HMGB3) also influence this pathway; TIPE promotes NSCLC cell proliferation by downregulating Wnt/β-catenin signaling, while HMGB3 is linked to poor prognosis due to its role in enhancing Wnt signaling activity [54,55]. 

Targeting the Wnt/β-catenin pathway in lung cancer offers a potential strategy to overcome resistance to targeted therapies like EGFR-TKIs and ALK inhibitors. Dysregulation of this pathway, often through mechanisms such as FOXM1 stabilization, WIF1 methylation, or FLNA and ANXA2 activity, contributes to treatment resistance in NSCLC. Notably, alterations in Wnt/β-catenin signaling, including reduced circFBXW7 expression or secondary CTNNB1 mutations, have been linked to decreased efficacy of therapies like osimertinib and ALK TKIs. Strategies that modulate β-catenin stability or Wnt signaling activity, such as restoring circFBXW7 function, may resensitize resistant cancer cells and enhance therapeutic outcomes, underscoring the pathway’s significance in lung cancer management [54,55].

### 4.7. The VEGF/VEGFR Pathway

Vascular endothelial growth factor (VEGF), predominantly secreted by tumor cells, stromal cells, and endothelial cells within the tumor microenvironment (TME), consists of several family members, including VEGF-A, VEGF-B, and VEGF-C. These factors exert their effects by binding to specific receptors, mainly tyrosine kinase receptors (VEGFRs) and neuropilin receptors (NRPs) [56,57]. 

VEGF-A, the most important for angiogenesis, primarily binds to VEGFR-2, leading to receptor dimerization and the activation of downstream signaling pathways essential for endothelial cell proliferation and migration (Figure 10). When VEGF-A binds to VEGFR-2, it triggers several intracellular signaling cascades, including phospholipase C γ (PLC-γ) and phosphoinositide 3-kinase (PI3K). PLC-γ hydrolyzes phosphatidylinositol-4,5-bisphosphate to produce inositol triphosphate (IP3) and diacylglycerol (DAG), leading to an increase in intracellular calcium and the activation of protein kinase C (PKC). PKC, in turn, stimulates the Raf1-MEK1/2-ERK1/2 pathway, promoting endothelial cell proliferation. Concurrently, PI3K activation results in AKT phosphorylation, which enhances endothelial nitric oxide synthase (eNOS) activity, supporting cell proliferation and migration. These processes are critical for the angiogenic response and tumor growth [56,57].

In lung cancer, VEGF promotes the formation of new blood vessels, which are necessary for tumors to grow and metastasize. VEGF achieves this by binding to its primary receptor, VEGFR-2, on endothelial cells, activating pathways that promote cell proliferation and survival. In NSCLC, high VEGF levels are associated with poor prognosis, increased tumor recurrence, and metastasis. VEGF also directly impacts tumor cells by promoting their proliferation and survival. Bevacizumab, a treatment for lung cancer, inhibits VEGF-A activity, which is often upregulated by hypoxia-inducible factor 1 (HIF-1). This inhibition prevents the abnormal angiogenesis driven by onco-metabolic stress, making bevacizumab a cytostatic rather than a cytotoxic agent [58,59].

### 4.8. The NF-κB Pathway

Nuclear factor-kappa B is a critical transcription factor that regulates inflammation, cell survival, and proliferation. Its dysregulation is implicated in various cancers, including lung cancer. The NF-κB family consists of five proteins—p65 (RelA), RelB, c-Rel, p50/p105 (NF-κB1), and p52 (NF-κB2)—that form dimers and bind DNA to regulate gene expression. p65, RelB, and c-Rel act as transcriptional activators, while p50 and p52 primarily function as repressors. NF-κB controls numerous genes related to key cellular processes, making it a central player in lung cancer by modulating both inflammation and cell survival [60,61]. The canonical NF-κB pathway is primarily activated by the phosphorylation of IκB proteins, which are inhibitors of NF-κB, by the IκB kinase (IKK) complex. This phosphorylation leads to the proteasomal degradation of IκB, allowing the translocation of NF-κB dimers, such as p65/RelA and p50, into the nucleus, where they initiate the transcription of genes that promote tumor progression (Figure 11). Activation of NF-κB in NSCLC is frequently mediated by the PI3K/AKT pathway, which is triggered by growth factors, cytokines, or mutations in key oncogenes, such as KRAS and EGFR [61]. The NF-κB signaling pathway is constitutively activated in various solid tumors, including lung cancer. Both small-cell lung cancer and non-small-cell lung cancer demonstrate high levels of NF-κB activation, which correlates with disease progression, advanced TNM stages, and poor prognosis. 

Mechanistically, NF-κB activation in lung cancer can be oncogene-driven, such as by K-Ras mutations frequently found in lung adenocarcinomas. Mutant K-Ras collaborates with p53 loss to activate both canonical and non-canonical NF-κB pathways, enhancing tumor cell survival, proliferation, and resistance to apoptosis [61]. Inflammatory stimuli also play a key role: chronic inflammation from factors such as cigarette smoke contributes to NF-κB-mediated cytokine production, establishing a tumor-promoting microenvironment. Furthermore, NF-κB is implicated in lung tumor angiogenesis and metastasis by regulating angiogenic factors like VEGF and promoting epithelial-mesenchymal transition (EMT) through Twist-1 upregulation. Together, these mechanisms underscore the multifaceted role of NF-κB in lung cancer pathogenesis [61,62].

Given the central role of NF-κB in lung cancer, targeting this pathway presents a promising therapeutic avenue. Current strategies include proteasome inhibitors such as Bortezomib, which block NF-κB activation by preventing IκB degradation. Although monotherapy has shown limited efficacy in clinical trials, combinations with cytotoxic agents hold potential. Non-steroidal anti-inflammatory drugs (NSAIDs), including aspirin and sulindac, inhibit IKK activity and enhance chemotherapy-induced apoptosis in lung cancer cells. Natural compounds such as curcumin and resveratrol also exhibit NF-κB suppression by targeting multiple signaling pathways. Despite these advances, challenges remain due to NF-κB’s dual roles in tumor promotion and inhibition, emphasizing the need for precise targeting and careful clinical evaluation to maximize therapeutic benefits while minimizing adverse effects [62].

### 4.9. The PD-1/PD-L1 Pathway

The PD-1/PD-L1 pathway is a crucial immune checkpoint mechanism that regulates immune tolerance within the tumor microenvironment. PD-1, expressed on T cells, interacts with PD-L1, a ligand often overexpressed in tumors, resulting in suppressed T-cell activity, reduced cytokine production, and impaired immune surveillance. PD-L1 expression in NSCLC is intricately regulated by several signaling pathways, including EGFR-driven mechanisms. EGFR activation enhances PD-L1 transcription through three primary routes: (1) JAK2/STAT1 signaling directly promotes PD-L1 gene expression; (2) Ras-BRAF-MEK-ERK signaling activates transcription factors that upregulate PD-L1; and (3) c-Cbl-dependent activation of JNKs induces c-Jun-mediated PD-L1 transcription (Figure 12). This regulation complicates treatment but also offers therapeutic opportunities [63]. 

In NSCLC, PD-L1 expression correlates with advanced tumor stages and poorer prognoses. Silencing PD-L1 can mitigate these effects, increasing survival rates by restoring antitumor immune responses [63,64]. Importantly, the density of tumor-infiltrating lymphocytes (TILs) is inversely related to PD-L1 levels; higher TIL levels are associated with better patient outcomes, emphasizing the axis’s significance in cancer progression. However, the response to PD-1/PD-L1 blockade remains limited to a portion of patients, leaving many unresponsive to monotherapy. For NSCLC patients with high PD-L1 expression, Atezolizumab has been shown to significantly extend overall survival compared to platinum-based chemotherapy [65,66].

Combining EGFR inhibitors with PD-1/PD-L1 blockers has shown promise in reducing resistance and extending the efficacy of immunotherapy. As checkpoint inhibitors like nivolumab and pembrolizumab reawaken T-cell activity, their integration with targeted therapies could redefine treatment strategies, offering improved outcomes for patients with NSCLC [67].

### 4.10. The Apoptosis Pathway 

The apoptosis pathway involves two main mechanisms: the intrinsic and extrinsic pathways, which ultimately converge to activate caspases, the proteolytic enzymes responsible for executing cell death. The intrinsic pathway is triggered by cellular stress or damage, leading to mitochondrial outer membrane permeabilization (MOMP) via pro-apoptotic Bcl-2 family proteins like BAX and BAK. This results in the release of cytochrome c, which binds to APAF1 and dATP, forming the apoptosome that activates caspase-9. Caspase-9 then triggers executioner caspases (e.g., caspase-3), promoting DNA fragmentation and cellular disassembly [68]. Simultaneously, pro-apoptotic factors like SMAC neutralize XIAP, a protein that inhibits caspase activation. In contrast, the extrinsic pathway is initiated at the cell membrane through death receptors such as DR4/DR5, which are activated by ligands like TRAIL (Figure 13). This interaction forms the death-inducing signaling complex (DISC), comprising FADD and pro-caspase-8, resulting in caspase-8 activation. Caspase-8 either directly activates executioner caspases or cleaves BID to its truncated form (tBID), linking to the intrinsic pathway by facilitating mitochondrial cytochrome c release. Together, these pathways integrate signals to regulate cell death, maintaining tissue homeostasis and eliminating damaged cells [68,69].

In non-small-cell lung cancer, disruptions in these apoptotic pathways are common. Loss of key apoptotic factors like FasL and Apaf-1, as well as mutations in death receptors such as TRAIL receptor 2, can impair apoptosis and contribute to therapy resistance. Elevated levels of anti-apoptotic proteins, including Bcl-2 and pro-caspase-3, are frequently observed in NSCLC and are associated with poor prognosis and reduced treatment effectiveness [69,70]. Growth factor signaling pathways, such as those involving IGF-1R, EGFR, and K-Ras, exacerbate anti-apoptotic signaling in lung cancer, promoting cell survival and resistance to therapy. Additionally, deficiencies in MAPK pathways, such as JNK and p38, impair radiation-induced apoptosis. Elevated levels of MKP1/CL100 phosphatase may also impair JNK activity, further contributing to therapy resistance. Understanding these disruptions is crucial for developing targeted therapies to overcome resistance in NSCLC [69,70,71]. 

For lung cancer, targeted therapies aim to restore apoptotic signaling, which is crucial for overcoming resistance to conventional treatments and improving patient outcomes. TRAIL receptor agonists, such as dulanermin, activate the extrinsic apoptotic pathway by promoting the formation of the DISC complex [72]. In the intrinsic pathway, Bcl-2 inhibitors like venetoclax neutralize anti-apoptotic proteins, allowing the release of cytochrome c. SMAC mimetics, such as birinapant, and inhibit XIAP and other inhibitor of apoptosis proteins (IAPs), restoring caspase function. Additionally, therapies targeting pro-survival pathways, such as EGFR or ALK inhibitors, and radiation-sensitizers that target the MAPK pathway (e.g., JNK activators), can further enhance the apoptotic response in NSCLC. Clinical trials exploring combinations with standard chemotherapies, such as paclitaxel and carboplatin, or with targeted therapies like bevacizumab, show promise in improving the effectiveness of these treatments for advanced NSCLC [69,72].

### 4.11. The HIF (Hypoxia-Induced) Pathway

The HIF (hypoxia-induced) pathway plays a key role in cellular metabolism, especially in tumor biology. HIF is a transcription factor composed of α- and β-subunits, with HIF-1α playing a central role in oxygen-sensitive responses. Under normoxic conditions, HIF-1α is rapidly degraded through hydroxylation by prolyl hydroxylase enzymes (PHDs), which require oxygen and iron [73]. This process leads to its ubiquitination by the von Hippel-Lindau (VHL) protein and subsequent proteasomal degradation. In hypoxic conditions, PHD activity is impaired due to a lack of oxygen and necessary cofactors, allowing HIF-1α to stabilize and accumulate (Figure 14). Once stabilized, HIF-1α translocates to the nucleus, where it forms a complex with HIF-1β. This heterodimer binds to hypoxia response elements (HREs) in the promoter regions of target genes, activating the transcription of genes involved in glycolysis, angiogenesis (e.g., VEGF), and tumor survival. Additionally, non-hypoxic signals such as growth factors and oncogenic pathways can independently induce HIF-1α expression, further enhancing its activity in tumorigenesis [73,74,75].

The regulation of HIF-1α is influenced by both oxygen-dependent mechanisms, such as prolyl hydroxylase domain proteins (PHDs) and factor inhibiting HIF (FIH), and oxygen-independent mechanisms, including NF-κB, SP1, and STAT3. Together, these pathways contribute to the metabolic shift observed in tumors. In NSCLC, HIF-1α is overexpressed, driving glycolysis even in the presence of oxygen (the Warburg effect) and facilitating tumor adaptation to hypoxia. HIF-1α also enhances the expression of glycolytic enzymes and pro-angiogenic factors, promoting tumor progression and therapy resistance. Elevated HIF-1α levels are linked to poor prognosis in NSCLC, making it an important therapeutic target [75,76].

Targeting HIF-1α (hypoxia-inducible factor 1-alpha) offers a promising strategy for lung cancer treatment, as the HIF-1α pathway plays a key role in tumor adaptation to low oxygen, angiogenesis, metabolic changes, and survival, all contributing to cancer progression. Disrupting this pathway can slow tumor growth. Several therapeutic approaches focus on inhibiting various stages of HIF-1α, such as blocking mRNA expression, protein stability, dimerization, and DNA binding. Drugs like anthracyclines and aminoflavone inhibit HIF-1α mRNA expression, while mTOR inhibitors (rapamycin, everolimus) and certain microtubule-targeting agents block its synthesis [77,78]. CRLX-101, an innovative nanoparticle formulation, reduces HIF-1α protein stability, and HSP90 inhibitors, alongside antioxidants like vitamin C, promote its degradation. Agents such as cyclo-CLLFVY and acriflavine target HIF-1α dimerization and DNA binding, halting its transcriptional activity. Additionally, nanotechnology enhances the delivery and effectiveness of these therapies, with treatments like cisplatin combined with HIF-1 inhibitors improving outcomes by overcoming resistance. Novel siRNA-based therapies using nanoparticles to silence HIF-1α gene expression also hold potential for enhancing chemotherapy efficacy. Other treatments, such as hemoglobin-based nanocarriers, aim to regulate oxygen levels and inhibit angiogenesis, further modulating the tumor microenvironment [78].

### 4.12. The Notch Signaling Pathway 

The Notch signaling pathway is a critical regulator of various cellular processes, including proliferation, differentiation, and apoptosis, and its dysregulation is implicated in the progression of NSCLC. The pathway begins with the synthesis of Notch receptors, which undergo post-translational modifications in the endoplasmic reticulum (ER) and Golgi apparatus, leading to the production of heterodimeric receptors composed of the extracellular and intracellular domains. Upon ligand binding, a conformational change exposes the S2 cleavage site, which is processed by ADAM metalloproteases, triggering the release of a Notch extracellular truncation (NEXT) peptide and facilitating further cleavage by γ-secretase at the S3/S4 sites [79,80]. This results in the release of the Notch intracellular domain (NICD), which translocates to the nucleus and activates the transcription of target genes involved in tumorigenesis (Figure 15). 

In NSCLC, Notch1 overexpression, particularly under hypoxic conditions, promotes tumor growth by enhancing angiogenesis and metabolic reprogramming. However, the effects of Notch signaling are context-dependent, with different outcomes observed in various NSCLC cell types. For example, inhibition of Notch1 has been shown to induce apoptosis in some NSCLC cells, while in squamous cell carcinoma (SCC) it can promote cell survival and proliferation. Additionally, Notch signaling interacts with other key pathways, such as PI3K/AKT, mTOR, and β-catenin, further contributing to cancer progression [80,81]. 

The complexity of treating NSCLC arises from the presence of distinct mutations that affect therapeutic outcomes. Oncogenic drivers, such as EGFR, ALK, DDR1, KRAS, and Notch, each contribute to the development and progression of NSCLC, and play significant roles in therapeutic resistance. Notch signaling, in particular, can be irregularly expressed or mutated, often synergizing with other mutations to create a more aggressive phenotype and increasing therapeutic resistance. Various strategies have been explored to target Notch signaling in NSCLC, with the aim of overcoming these challenges. These include gamma-secretase inhibitors (GSIs), monoclonal antibodies against Notch receptors and ligands, alpha-secretase inhibitors targeting ADAMs, and stapled peptides to block Notch/Mastermind interactions [82]. GSIs, such as DAPT, PF-03084014, and RO4929097, are among the most investigated compounds, which work by inhibiting the γ-secretase enzyme involved in Notch receptor cleavage and NICD production. Additionally, monoclonal antibodies targeting specific Notch ligands, like enoticumab and demcizumab, as well as inhibitors of other key components like ADAMs and Nicastrin, have been tested in preclinical models and clinical trials. These approaches aim to disrupt the Notch signaling cascade, potentially enhancing the efficacy of existing therapies and providing new avenues for treatment in NSCLC and other malignancies [82,83].

## 5. Genomic Alterations and Instabilities in Lung Cancer

Lung cancer is driven by various genomic alterations, including point mutations, insertions/deletions, copy number variations, gene fusions, and rearrangements, all of which contribute to tumor initiation and progression. These mutations frequently affect key oncogenes and tumor suppressor genes, leading to dysregulated processes like uncontrolled proliferation, evasion of apoptosis, and increased metastatic potential [84].

KRAS mutations, especially at codon 12 (e.g., G12C, G12D, G12V), result in the continuous activation of the RAS signaling pathway, driving uncontrolled proliferation. Mutations like KRAS G12D are associated with poorer clinical outcomes, including shorter progression-free survival (PFS) and overall survival (OS). In contrast, nucleotide substitutions, such as guanine to thymine (KRAS G>T), have been associated with more favorable outcomes, suggesting potential prognostic differences depending on amino acid substitution. Research indicates that patients with KRAS G>T substitutions may have improved progression-free and overall survival compared to those with G>C or G>A substitutions, possibly reflecting distinct biological behaviors in KRAS-driven lung cancers [84,85].

EGFR mutations, mainly in exons 18–21, affect the tyrosine kinase domain, with around 90% involving microdeletions in exons 19 and 21. These mutations make tumors highly responsive to EGFR tyrosine kinase inhibitors (TKIs), although their role in surgically resectable lung cancer is still debated [85].

ALK gene mutations, such as His694Arg (H694R) and Glu1384Lys (E1384K), activate downstream pathways like STAT3 and AKT, promoting oncogenesis. This has made ALK inhibitors like Lorlatinib, Ceritinib, and Brigatinib essential for treating ALK-positive tumors [86,87].

In addition to specific mutations, genomic instability—particularly chromosomal instability (CIN) and microsatellite instability (MSI)—plays a key role in lung cancer progression:
CIN contributes to chromosomal aberrations such as gain of chromosome 5p (enhancing TERT expression for tumor growth) or loss of chromosome 9p21 (inactivating CDKN2A, leading to unchecked proliferation). CIN promotes karyotype heterogeneity, fostering genetic diversity within tumors, complicating treatment, and enabling adaptation to evolving conditions. CIN also supports the creation of an inflammatory tumor microenvironment that promotes pro-tumor signaling and metabolic changes that favor cancer cell survival and immune evasion [88].MSI is less common in lung cancer, but when present, it often arises due to defects in DNA mismatch repair (MMR) genes like MSH2 and MLH1, leading to a hypermutated state and increased neoantigen formation, making MSI-high tumors more responsive to immunotherapy. MSI mutations, such as those in TGFBR2, further promote genomic instability and cancer progression [89].

## 6. **Genetic Testing and Biomarkers in Lung Cancer**

One of the most significant advancements in lung cancer diagnosis and treatment over the past decade has been the transition towards personalized medicine. In this approach, therapeutic decisions are guided by the unique histologic and genetic characteristics of the patient’s tumor, allowing for more targeted treatment strategies [90]. Genetic testing is now a standard practice for patients with NSCLC. 

In lung cancer, the detection of specific molecular changes helps classify tumors more accurately, guiding clinicians in determining the most appropriate course of treatment. By identifying these genetic alterations, oncologists can prescribe therapies that target the molecular mechanisms driving cancer growth [91]. This approach not only improves treatment efficacy but also reduces unnecessary interventions, thereby minimizing the potential side effects of broader therapies. Drawing on the Centers for Disease Control Genetics Working Group framework, the applicability of genetic testing for lung cancer prevention can be evaluated based on several critical criteria. Notably, a high proportion of the population carries genetic variants linked to an increased risk of developing lung cancer, suggesting a broad potential for genetic testing. 

However, there are significant challenges, including the need to maintain high-quality standards in genetic testing laboratories and the costs associated with widespread genetic testing. Furthermore, the strength of the association between the identified genotype and lung cancer risk has not been prospectively validated, raising concerns about the clinical utility of these tests. Therefore, while genetic testing holds promise, further research and refinement are needed to meet the rigorous standards required for its effective use in lung cancer prevention [92].

Lung cancer screening, as recommended by the National Comprehensive Cancer Network (NCCN), provides several significant benefits (Figure 16). Early detection of lung cancer can reduce mortality by enabling timely interventions that improve survival rates. Patients diagnosed at earlier stages often require less aggressive treatment, which helps to reduce the side effects of both the disease and its treatments [93]. Additionally, lung cancer screening can motivate healthier lifestyle changes, such as smoking cessation, and reduce the psychosocial burden by relieving anxiety related to lung cancer risk. Screening also helps to identify tumors when they are smaller and less invasive, contributing to a better overall quality of life for patients.

Despite its advantages, genetic testing in the context of lung cancer carries certain risks. One concern is the potential to identify indolent tumors—those that may not cause harm during the patient’s lifetime—which could lead to unnecessary treatment, potentially reducing quality of life. The psychological impact of test results, including anxiety and stress, is also a key consideration. There are physical risks associated with follow-up diagnostic procedures, such as biopsies, and the possibility of false-positive or false-negative results. False positives may lead to unnecessary treatments, while false negatives could result in missed opportunities for early intervention. Furthermore, the cost of genetic testing and subsequent follow-up procedures, including radiation exposure during diagnostic imaging, adds an additional layer of complexity.

Next-generation sequencing (NGS) has revolutionized lung cancer diagnostics and treatment strategies. In the era of precision medicine, this technology enables comprehensive tumor molecular profiling, providing critical insights into a wide range of cancer-related mutations. This increases the possibility of identifying germline mutations that may have implications for both treatment and prevention [94]. NGS facilitates a comprehensive examination of the genome or exome in any type of cancer by enabling the parallel sequencing of millions of short reads from various nucleic acids, such as micro-RNA and other non-protein-coding DNA species [95]. NGS applications include whole-genome sequencing (WGS), whole-exome sequencing (WES), RNA expression profiling, and targeted oncology panels that may cover anywhere from a few to hundreds of genes. 

While NGS offers significant advantages that have made it indispensable in both research and clinical practice, it also presents certain limitations, notably the requirement for sophisticated bioinformatics tools and highly trained personnel to manage the experimental procedures and data analysis. Some advantages of NGS include lower costs, quick processing times, a wide range of applications, and utility in both research and clinical environments [96]. Additionally, commercial NGS platforms and specialized kits are widely available. However, NGS also has disadvantages, such as the need for specialized software and computing resources for data analysis, a lack of standardized protocols or materials for clinical applications, and remaining cost barriers in certain developing countries. Beyond NGS, several other genetic testing techniques are crucial in lung cancer diagnostics. Fluorescence in situ hybridization (FISH) can identify chromosomal abnormalities when traditional sequencing methods fail, making it a valuable complementary technique, especially in cases where limited tumor tissue is available. Additionally, polymerase chain reaction (PCR) remains a gold standard for detecting specific mutations in genes like EGFR, KRAS, and BRAF [96,97].

Building upon the advancements of NGS in evaluating intratumoral genetic heterogeneity (ITGH) in NSCLC, recent studies have identified the limitations of short-read sequencing in fully characterizing structural variants (SVs) within tumor genomes. NGS, which is primarily effective in detecting single nucleotide variants (SNVs) and copy number variants (CNVs), often relies on fragmented DNA sequences amplified through PCR, which can hinder its ability to identify larger SVs in complex or repetitive regions. Although WGS has offered a broader lens by examining multiregional tumors, its reliance on short reads constrains its sensitivity for detecting certain structural alterations [98,99].

Optical genome mapping (OGM) emerges as a powerful complementary method, addressing these limitations by analyzing ultra-high molecular weight (UHMW) DNA without the biases introduced by DNA fragmentation and amplification [98]. Using fluorescent markers to label long, intact DNA molecules, OGM provides a high-resolution view of SVs, especially those exceeding 5 kilobases, which are often undetected by WGS [100]. This technique allows for the de novo identification of SVs across primary and metastatic sites, revealing previously uncharacterized SVs that contribute significantly to ITGH. Combining WGS and OGM enables a comprehensive approach that captures a wide spectrum of genetic diversity within NSCLC tumors, enhancing our understanding of the role of large SVs in tumor progression and metastasis. Notably, OGM has proven particularly informative in identifying private SVs specific to metastases, elucidating the complex interplay between primary tumors and metastatic niches, and uncovering insights into the genetic drivers underlying metastatic potential. Thus, OGM offers a crucial tool in cancer genomics, providing insights that enhance both diagnostic precision and the development of targeted therapeutic strategies [99,100,101].

Emerging technologies like liquid biopsy have gained considerable attention in recent years due to their minimal invasiveness and ability to provide continuous, real-time insights into tumor biology [102]. This technique allows for repeated sampling of biological material such as circulating tumor cells (CTCs), extracellular vesicles (EVs), cell-free DNA (cfDNA), circulating tumor DNA (ctDNA), non-coding RNAs (ncRNAs), and proteins, among others [103]. 

These biomarkers, detectable in bodily fluids such as blood, cerebrospinal fluid, saliva, pleural fluid, and urine, offer essential information for the early diagnosis of lung cancer, therapeutic monitoring, and prognostic assessment. The concentrations of these tumor markers can vary depending on factors like tumor stage, metastasis, and genomic instability, making them valuable for tracking tumor progression and therapy response [104,105].

Circulating free DNA (cfDNA) consists of fragmented DNA released into the bloodstream through natural processes like apoptosis, necrosis, or active secretion from both healthy and damaged cells, including those from tumors. This fragmented DNA, often derived from multiple tissues, plays a pivotal role as a non-invasive biomarker in cancer research. In NSCLC, elevated cfDNA levels have been associated with a poorer prognosis, especially when linked to additional markers like the neutrophil-to-lymphocyte ratio (NLR) [106,107].

Circulating tumor DNA (ctDNA), a subset of circulating free DNA (cfDNA), offers significant promise as a tumor biomarker due to its ability to provide real-time insights into the tumor’s genetic makeup. ctDNA is released into the bloodstream primarily through apoptosis and necrosis of tumor cells and represents only a small fraction of the total cfDNA, typically ranging from 0.01% to 90% of cfDNA in peripheral blood [59]. Its unique properties, such as a short half-life of approximately 15 min to 2.5 h, make ctDNA a dynamic indicator of tumor activity, as it can reflect changes in tumor status much earlier than traditional protein markers, which may take weeks to appear [108]. 

The genetic content of ctDNA includes vital information on tumor mutations, methylation patterns, and microsatellite instability, allowing for detailed genomic analyses that can support early diagnosis, monitor disease progression, and track the emergence of drug resistance in cancer patients. For instance, ctDNA has proven to be particularly valuable in non-small-cell lung cancer, where it helps detect EGFR mutations, enabling clinicians to tailor targeted therapies more effectively and monitor patients’ responses to treatment [109,110].

The clinical utility of ctDNA is further enhanced by advanced detection techniques such as digital PCR, quantitative PCR (qPCR), and next-generation sequencing, which offer high sensitivity and specificity. These methods allow for both qualitative and quantitative analysis of ctDNA, even when the circulating levels are low, which is a common challenge in early-stage lung cancer and minimal residual disease settings [110]. Moreover, ctDNA testing is less influenced by tumor heterogeneity than tissue biopsies, making it a more reliable tool for comprehensive genomic profiling. In lung cancer, ctDNA-based liquid biopsies enable the detection of emerging genetic mutations related to drug resistance, offering critical insights for modifying treatment plans. 

Despite its low abundance in blood, ctDNA’s association with tumor burden has been well established, with studies showing elevated levels correlating with advanced cancer stages. The ability of ctDNA to predict recurrence and metastasis, alongside its application in non-invasive liquid biopsy technologies, positions it as a next-generation biomarker for early diagnosis, prognosis, and real-time monitoring of tumor dynamics [111,112].

Circulating tumor cells (CTCs) have emerged as a significant biomarker in the context of liquid biopsy for non-small-cell lung cancer (NSCLC), providing valuable insights into disease progression and treatment response. CTCs, alongside other liquid biopsy analytes like circulating tumor DNA and extracellular vesicles, offer a non-invasive method for monitoring cancer and detecting therapeutic resistance mechanisms. Notably, CTCs have been shown to predict patient outcomes, with specific markers like PD-L1+ CTCs being linked to poorer progression-free and overall survival. This biomarker also aids in assessing resistance to immunotherapy. While ctDNA is useful for identifying mutations and guiding targeted therapies, CTCs offer a more comprehensive analysis by representing whole cells, making them a more reliable indicator of tumor behavior, especially when combined with PD-L1+ EV analysis. Further research is necessary to standardize liquid biopsy procedures and confirm these findings, but the integration of multiple liquid biopsy markers, including CTCs, holds promise for improving personalized cancer treatment strategies [113,114]. 

The research on CTCs and angiogenesis highlights their critical roles in cancer progression and metastasis. Angiogenesis, regulated by factors like VEGF, leads to the formation of leaky, immature blood vessels, facilitating CTC intravasation and aiding metastasis. Studies show that higher microvessel density in tumors correlates with increased metastasis risk, particularly in cancers like breast cancer. CTCs, which can form clusters with enhanced metastatic potential, are abundant in aggressive cancers such as small-cell lung cancer (SCLC), where they contribute to the rapid spread of the disease. Despite advancements in detecting CTCs, their clinical utility as biomarkers is limited due to detection challenges, lack of standardization, and their heterogeneity. In cancers such as non-small-cell lung cancer, where CTC counts are typically lower, the prognostic value of these cells remains unclear. Moreover, while CTCs offer valuable insights into tumor biology, their use in guiding therapy lags behind other biomarkers like cell-free DNA. Overall, the study of CTCs continues to evolve, but more standardized approaches are needed to fully harness their potential in clinical practice [114,115].

Extracellular vesicles (EVs) are small, lipid bilayer-bound vesicles secreted by cells, involved in intercellular communication and the regulation of tumor processes. EVs, including exosomes and microvesicles, carry genetic materials such as miRNAs, lncRNAs, and proteins, making them vital for tumor progression and immune modulation. In lung cancer, EVs are emerging as promising liquid biopsy tools due to their stability and capacity to reflect real-time tumor dynamics [116,117]. Specific miRNAs, like miR-934, miR-186-5p, miR-29a-3p, and miR-497-5p, carried by EVs have been linked to lung cancer progression. Additionally, EV-derived circRNAs such as circUSP7 and circSATB2 and lncRNAs like ZEB2-AS1 and UFC1 have shown significant potential for early lung cancer detection. These biomarkers, by regulating various signaling pathways, offer promising avenues for disease diagnosis and monitoring. Despite their diagnostic promise, the low abundance of EVs in biological samples presents a challenge, necessitating the refinement of current isolation and characterization techniques to improve clinical applications [116,117]. 

The versatility of liquid biopsy lies in its ability to detect a wide range of molecular markers, including epigenetically modified DNA, long non-coding RNAs, microRNAs, tumor-associated antigens, and metabolites [117]. These components not only provide critical insights into the genetic and molecular characteristics of the tumor but also help identify mutations in tumor-associated genes, enabling personalized treatment strategies in precision medicine. In clinical practice, liquid biopsies are proving highly effective in monitoring disease recurrence and guiding treatment decisions, positioning them as a promising alternative to traditional tissue biopsies. As research progresses, the use of liquid biopsy is expected to expand, further enhancing lung cancer management through improved early detection and personalized therapeutic interventions.

## 7. Conclusions

Lung cancer continues to be a leading cause of cancer mortality globally, driven by a combination of environmental exposures and complex genetic alterations. Recent advancements in understanding its molecular underpinnings have shed light on the pivotal role of oncogenes and tumor suppressor genes. These discoveries have transformed the diagnostic landscape, enabling the use of precision medicine through targeted therapies that inhibit specific molecular pathways involved in tumor progression. Techniques such as next-generation sequencing and liquid biopsies now allow for real-time monitoring of genetic mutations, providing valuable insights into the disease’s behavior and guiding more personalized treatment approaches. 

These innovations have contributed significantly to improving early diagnosis and prognostic evaluations, particularly in cases of non-small-cell lung cancer. The expanding molecular understanding not only helps refine prognosis but also paves the way for more effective interventions. However, despite these significant strides, the treatment of lung cancer faces ongoing challenges, particularly in overcoming drug resistance, tumor heterogeneity, and the identification of actionable mutations in all patients. The integration of biomarkers in clinical practice has improved early diagnosis and treatment outcomes. While targeted therapies and immunotherapies have shown promise, their efficacy is often hampered by the development of resistance mechanisms, necessitating further research into combinatorial treatment strategies. 

Future directions in lung cancer management will likely prioritize refining personalized therapies by exploring novel molecular targets and enhancing the effectiveness of existing treatments. Expanding the use of real-time monitoring through genetic and liquid biopsy technologies will enable dynamic adjustments to treatment plans, improving both efficacy and timing. This approach is expected to allow for more precise tailoring of therapies to individual patient profiles, potentially reducing side effects and increasing survival rates. At the same time, efforts should focus on developing combinatory therapeutic strategies that target multiple signaling pathways and molecular mechanisms simultaneously. These multifaceted approaches may provide a more durable and sustained therapeutic effect. Continued innovation in genetic testing and molecular diagnostics will be crucial for guiding precision medicine, ensuring that therapies remain adaptable to the evolving nature of cancer. Ultimately, integrating molecular advancements with therapeutic innovation holds great promise for significantly improving long-term outcomes and enhancing the quality of life for lung cancer patients.

## Figures and Tables

**Figure 1 cancers-16-04048-f001:**
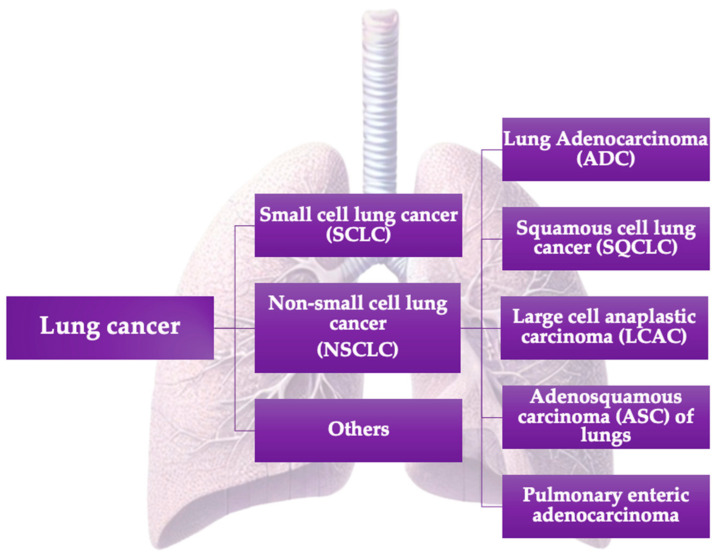
Classification of lung cancers.

**Figure 2 cancers-16-04048-f002:**
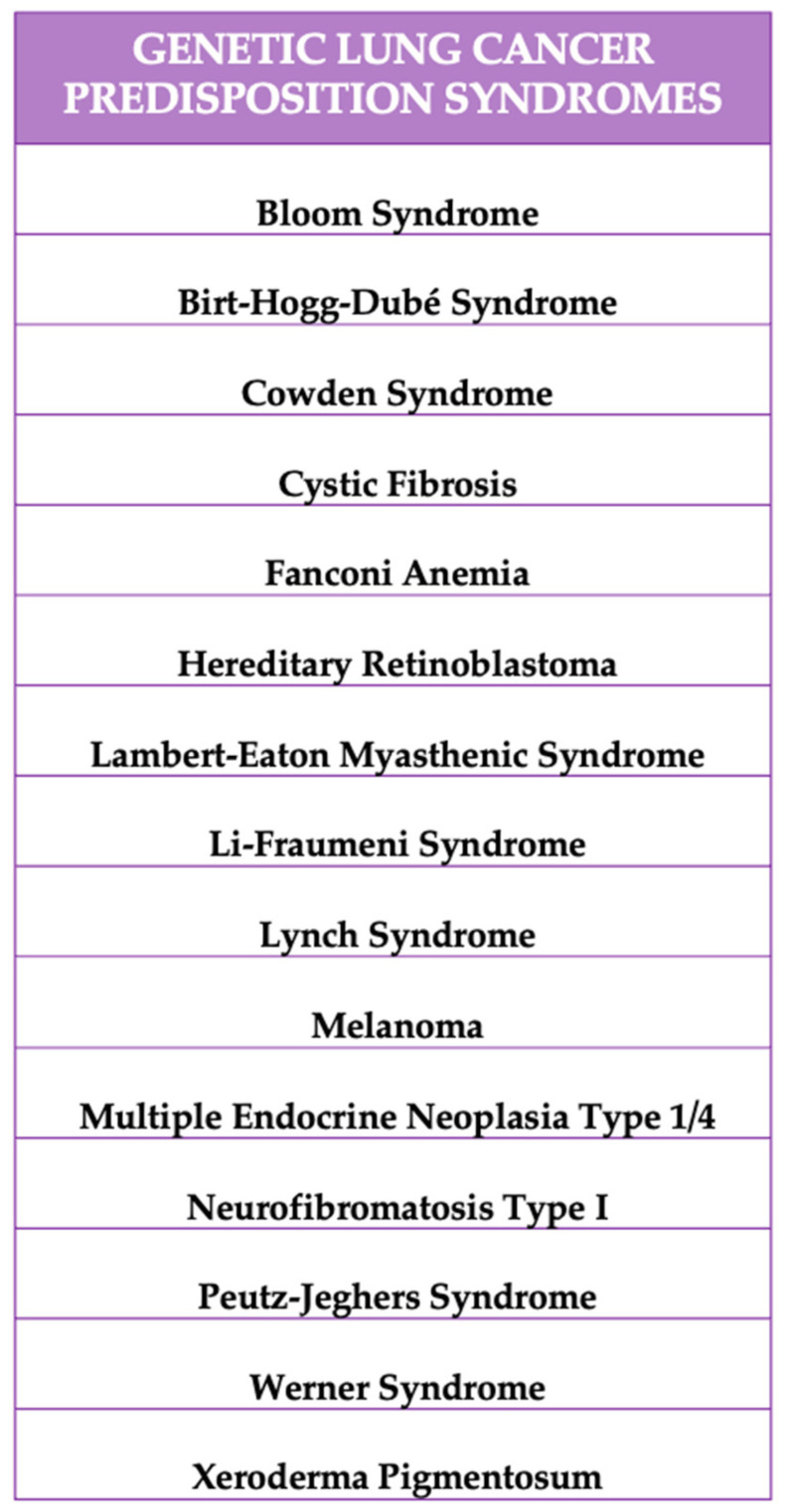
Lung cancer predisposition syndromes.

**Figure 3 cancers-16-04048-f003:**
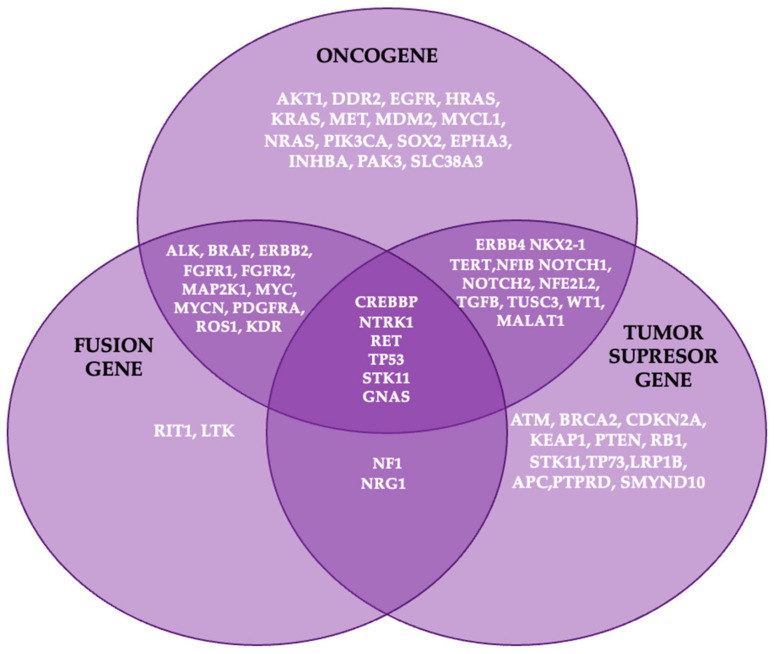
Genes involved in lung carcinogenesis.

**Figure 4 cancers-16-04048-f004:**
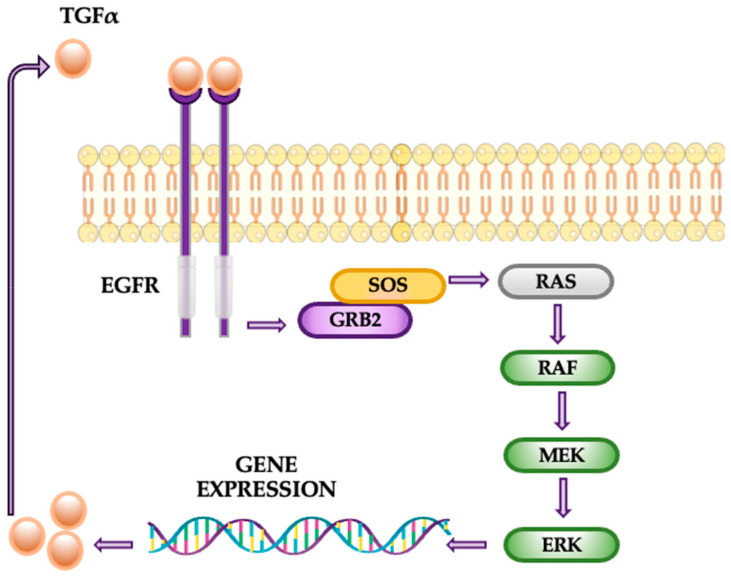
The MAPK pathway.

**Figure 5 cancers-16-04048-f005:**
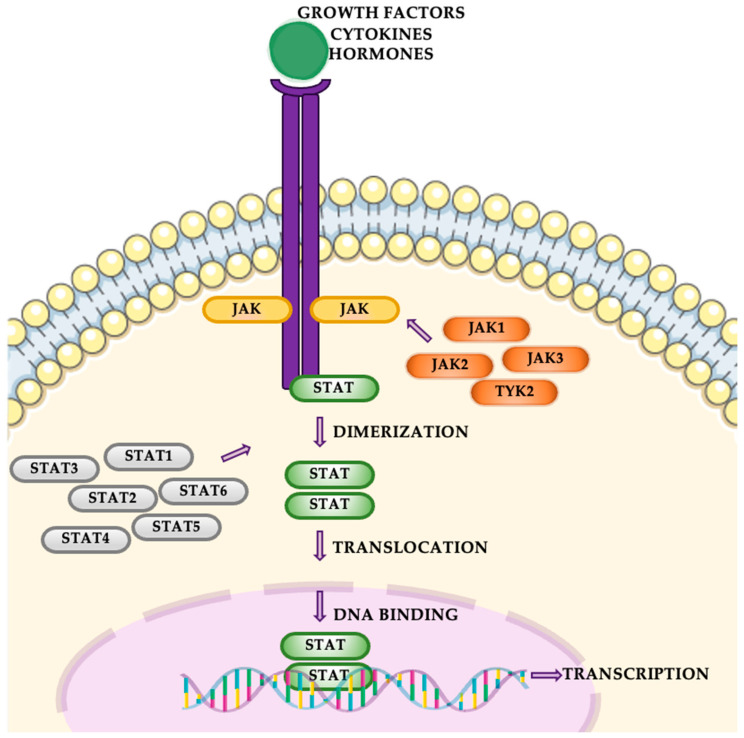
The JAK-STAT pathway.

**Figure 6 cancers-16-04048-f006:**
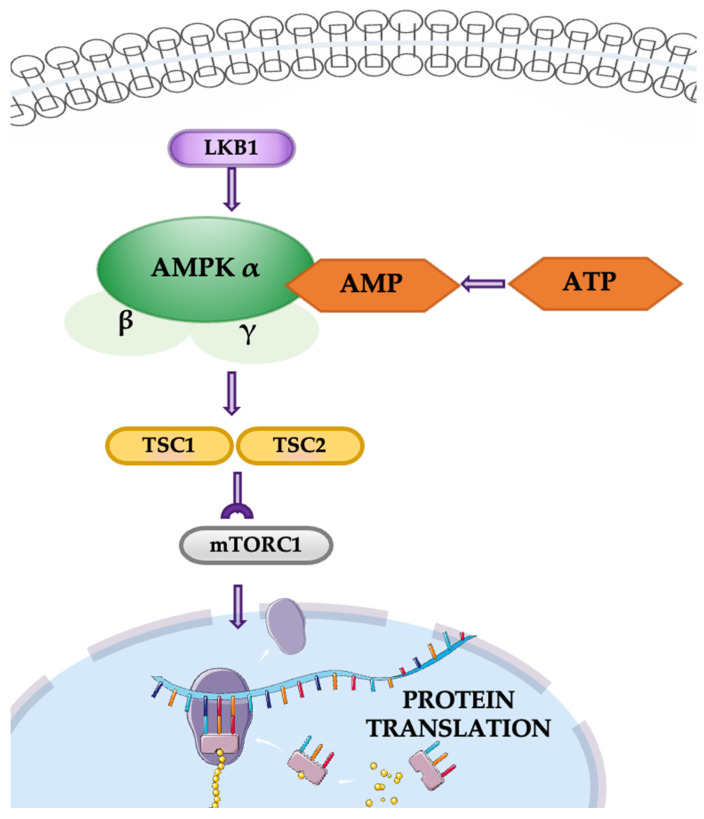
The LKB1/AMPK/mTOR pathway.

**Figure 7 cancers-16-04048-f007:**
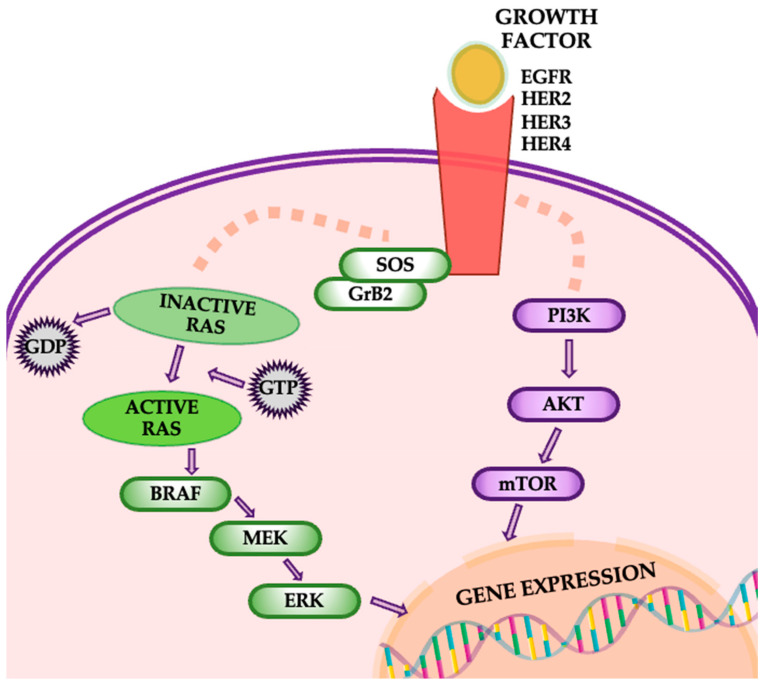
The EGFR/PI3K/Akt/mTOR pathway.

**Figure 8 cancers-16-04048-f008:**
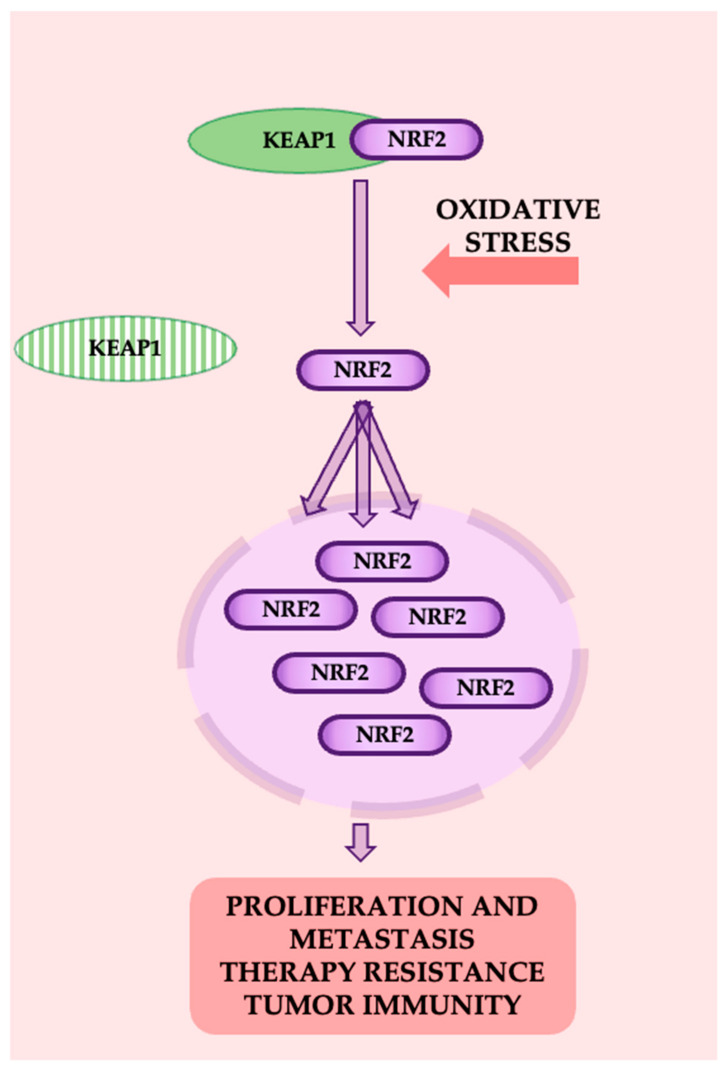
The Keap1-Nrf2 pathway.

**Figure 9 cancers-16-04048-f009:**
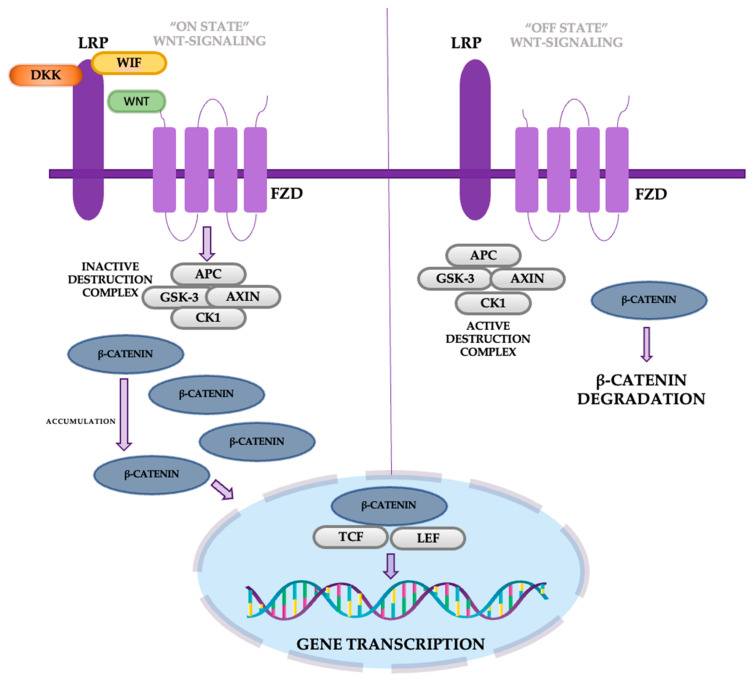
The Wnt/β-catenin pathway.

**Figure 10 cancers-16-04048-f010:**
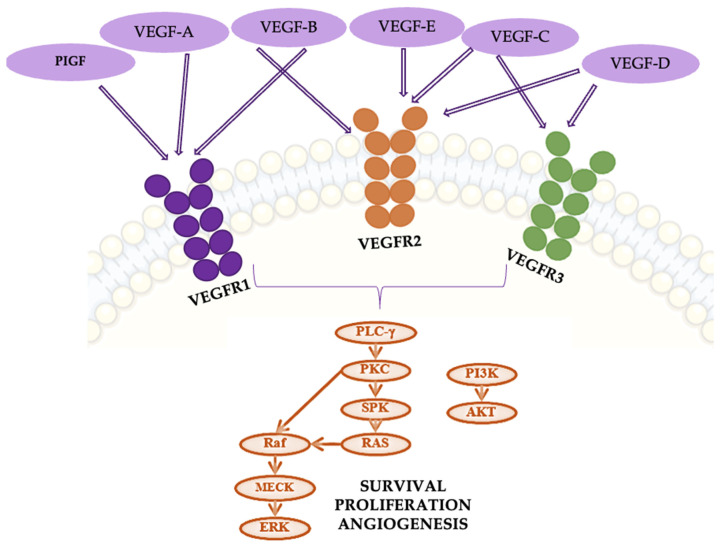
The VEGF/VEGFR pathway.

**Figure 11 cancers-16-04048-f011:**
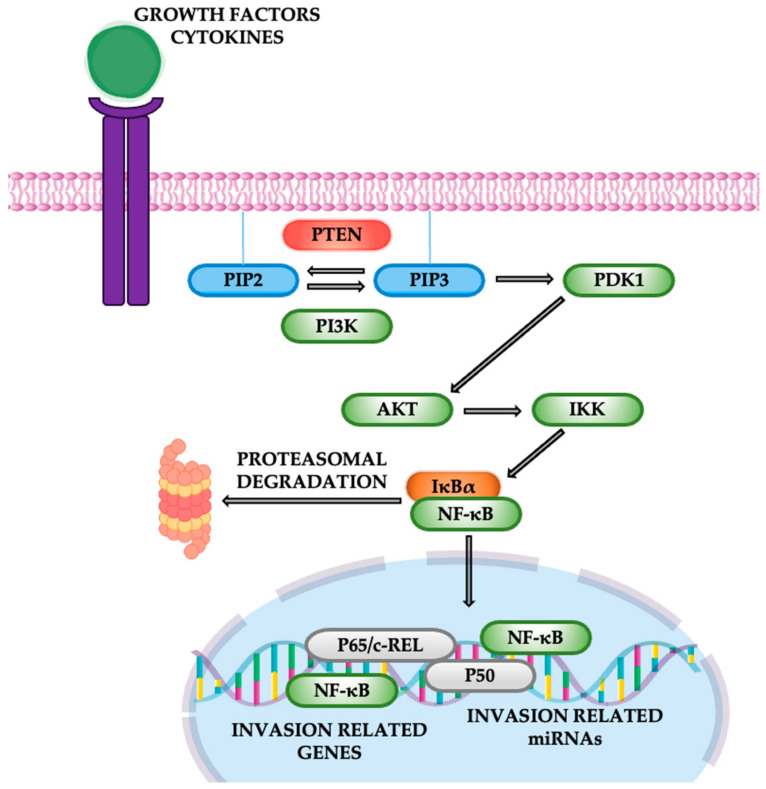
The NF-κB pathway.

**Figure 12 cancers-16-04048-f012:**
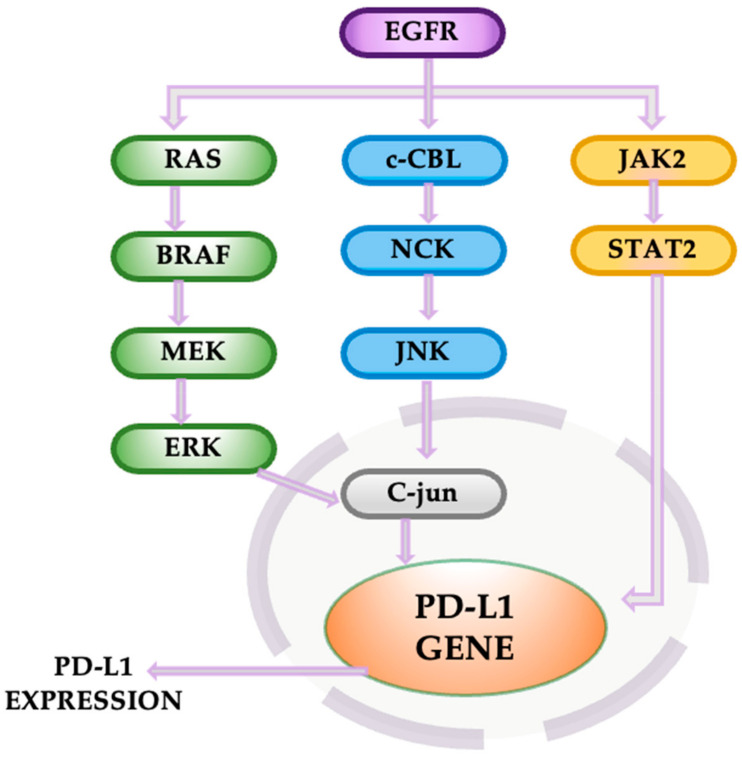
The PD-1/PD-L1 pathway in lung cancer.

**Figure 13 cancers-16-04048-f013:**
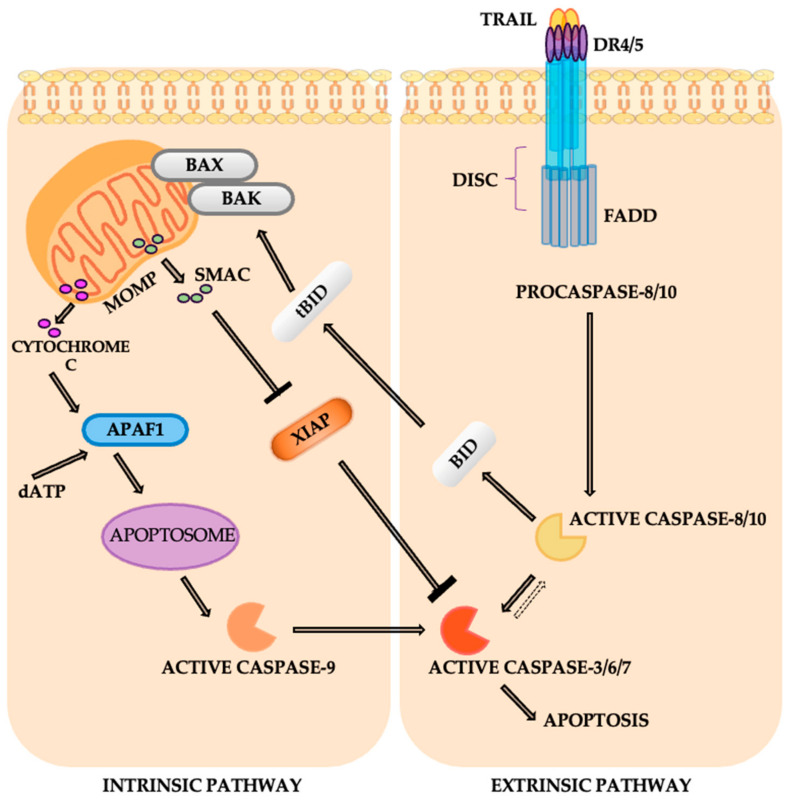
The apoptosis pathway.

**Figure 14 cancers-16-04048-f014:**
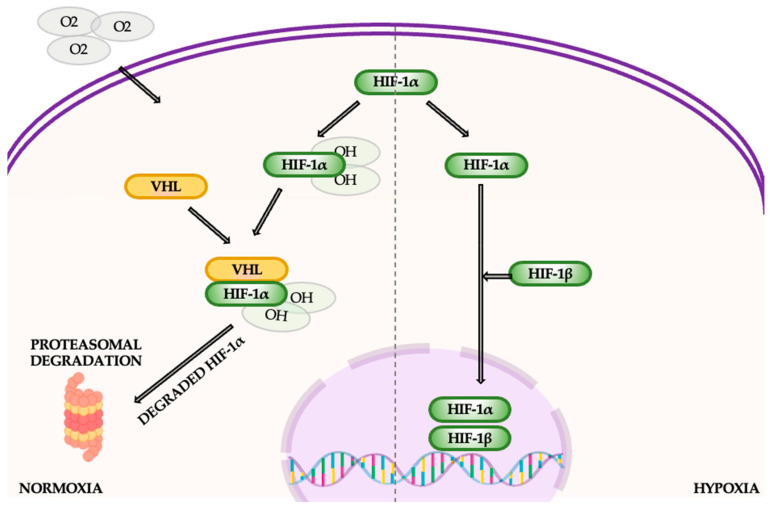
The hypoxia-induced pathway.

**Figure 15 cancers-16-04048-f015:**
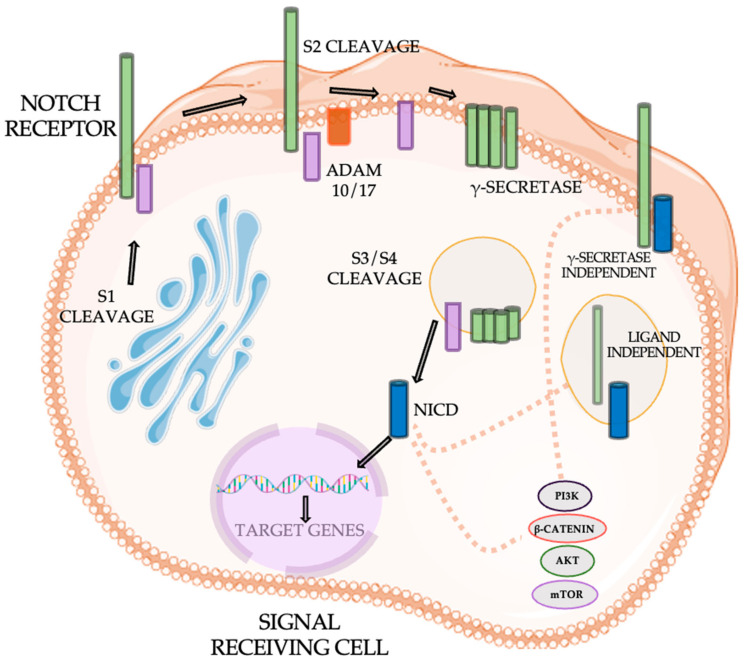
Notch pathway.

**Figure 16 cancers-16-04048-f016:**
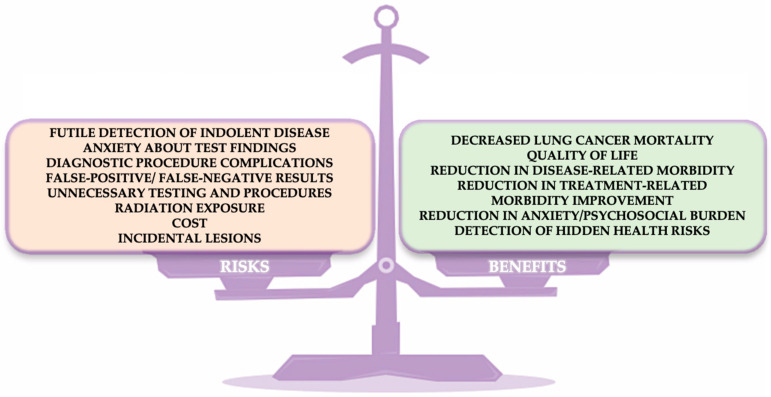
Benefits and risks of genetic testing.

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
