# Peer review of "Genetic Blueprints in Lung Cancer: Foundations for Targeted Therapies"

_cancers, 2024, doi:10.3390/cancers16234048_

Round 1
Reviewer 1 Report
Comments and Suggestions for Authors
In this manuscript,the authors systematically review lung cancer therapies from a genetic perspective,which provides a new way to cure lung cancer. This team also introduce both molecular pathways and genomic alterations in lung cancer,which are helpful for us to understand lung cancer. I really appreciate the work of this review, but there are still some parts to improve for the quality of the manuscript as follows.
General comments:
1、For this manuscript, authors should pay more attention to the comprehensibility of the figures , you should better adjust the font size and color so that you can improve the accuracy and readability.
2、The mechanism figure of molecular pathways lack logicality and does not match the content described in the paragraph.
3、In the introduction,authors discussed lung cancer situation in present society,but your manuscript topic--lung cancer genetic therapy—did not mentioned in previous part. Could you please add research status of lung cancer genetic therapy in your previous part.
Comments:
1、For the line 210 in page 8,‘plays a important role’should be changed to‘plays an important role’.
2、 For the line 314 in page 11,reference [47] was mistakenly inserted into the punctuation‘,’.
3、 For the line 384 in page 13,what does‘KRAS G>T’mean.
4、Fonts in Figure 14 should be changed because them cannot be recognized immediately.
5、PD-1/PD-L1 Pathway lacks mechanism figure,please add it in previous part.
Author Response
We greatly appreciate your thorough review and constructive feedback, which have been invaluable in enhancing the quality and clarity of our manuscript. Below, we have provided detailed responses to each of your comments and described the revisions made to address them comprehensively.
- For the line 210 in page 8,‘plays a important role’should be changed to‘plays an important role’
The grammatical issue has been corrected, and "plays a important role" now reads as "plays an important role."
- For the line 314 in page 11,reference [47] was mistakenly inserted into the punctuation‘,’.
The misplacement of reference [47] within the punctuation has been rectified.
- For the line 384 in page 13,what does‘KRAS G>T’mean.
The meaning of "KRAS G>T" has been clarified and is now elaborated upon in Line 584, Page 20.
- Fonts in Figure 14 should be changed because them cannot be recognized immediately.
We have revised the fonts in all figures for improved readability, using the "Palatino" font with sizes ranging between 7 and 10 for consistency and clarity.
- PD-1/PD-L1 Pathway lacks mechanism figure,please add it in previous part.
In response to your suggestion, we have added a detailed figure illustrating the PD-1/PD-L1 pathway and included corresponding textual explanations in the manuscript.
In addition to addressing these specific issues, we conducted a thorough review of the manuscript, correcting errors throughout and improving the overall presentation and accuracy.
General Comments
Figure Comprehensibility
1.To enhance figure comprehensibility, we adjusted font sizes, colors, and overall design to ensure greater accuracy and readability. Each figure now provides a clear and visually cohesive summary of the relevant pathways.
2.Logical Consistency of Mechanism Figures
We meticulously reviewed each molecular pathway figure and corresponding text for logical coherence. Necessary revisions were made to both figures and content to align with the manuscript’s narrative. We aimed to create visual abstracts of the molecular pathways, ensuring clarity and precision. Specific Revisions to Molecular Pathway Sections:
4.1. The MAPK Pathway: Revised text and replaced the figure with an updated version.
4.2. The JAK-STAT Pathway: Expanded textual details and enhanced figure clarity with additional elements.
4.3. The LKB1/AMPK/mTOR Pathway: Updated both the text and figure for improved logical flow and detail.
4.4. The EGFR/PI3K/Akt/mTOR Pathway: Revised the text and replaced the figure with a more informative depiction.
4.5. The Keap1-Nrf2 Pathway: Refined the text and enhanced the figure for greater visual impact.
4.6. The Wnt/β-catenin Pathway: Added more details to the text and enriched the figure with additional insights.
4.7. The VEGF/VEGFR Pathway: Adjusted figure font; the text remained unchanged.
4.8. The NF-κB Pathway: Significantly revised the text and introduced a more detailed figure.
4.9. The PD-1/PD-L1 Pathway: Expanded the text and included a new figure to address your request.
4.10. The Apoptosis Pathway: Updated the text and introduced a corresponding figure for clarity.
4.11. The HIF Pathway: Revised text details and replaced the figure for consistency.
4.12. The Notch Signaling Pathway: Enhanced both the textual explanation and the figure.
- Introduction—Lung Cancer Genetic Therapy: In response to your suggestion, we have expanded the introduction to include a comprehensive discussion of lung cancer genetic therapy. (Lines 43-59)
We trust these revisions address your concerns effectively and contribute to a more robust and impactful manuscript. Thank you once again for your insightful comments and for providing us with the opportunity to improve our work.

Reviewer 2 Report
Comments and Suggestions for Authors
Andra Dan and co-authors presented a comprehensive review of signaling pathways involved in lung cancer as possible targets for targeted therapy of various kinds. The idea to write such a review is still relevant. The knowledge of dysregulated signaling in lung cancer is important for both basic researchers and clinicians. The structure of the review is acceptable.
Unfortunately, the manuscript in its present form is not ready for publication.
There are many errors. For example, in chapter 4.1 MAPK there is a scheme of Stat pathway named as MAPK pathway. And vice versa, MAPK kinase signaling is named as Stat pathway (Figure 5).
The scheme of EGFR signaling (Figure 7) should include RAS and MAP kinases as classical responders for EGFR signaling.
In general, all schemes are quite simple and it is highly recommended to redesign them.
The manuscript should be proofread for typos.
Author Response
We are deeply grateful for your insightful comments, which have greatly contributed to refining the clarity and accuracy of our manuscript. Below, we provide detailed responses to each of your observations and outline the corresponding revisions made in the manuscript.
- For example, in chapter 4.1 MAPK there is a scheme of Stat pathway named as MAPK pathway. And vice versa, MAPK kinase signaling is named as Stat pathway (Figure 5).
We acknowledge the inadvertent error in labeling the MAPK pathway as the STAT pathway and vice versa in Figure 5. This has been corrected, and the text and figures have been carefully reviewed to ensure accuracy throughout the manuscript. Any additional inconsistencies identified during this review have also been rectified.
- The scheme of EGFR signaling (Figure 7) should include RAS and MAP kinases as classical responders for EGFR signaling.
In response to your observation, the figure representing EGFR signaling has been updated to include RAS and MAP kinases as classical responders for EGFR signaling. This addition aligns the visual representation with established pathway mechanisms and complements the textual description.
- In general, all schemes are quite simple and it is highly recommended to redesign them.
We greatly appreciate your feedback on the simplicity of the pathway figures. To address this, we have redesigned all figures for greater detail and visual coherence. Necessary revisions were made to both figures and content to align with the manuscript’s narrative. Below is a summary of the specific revisions made to the molecular pathway sections:
4.1. The MAPK Pathway: Revised text and replaced the figure with an updated version.
4.2. The JAK-STAT Pathway: Expanded textual details and enhanced figure clarity with additional elements.
4.3. The LKB1/AMPK/mTOR Pathway: Updated both the text and figure for improved logical flow and detail.
4.4. The EGFR/PI3K/Akt/mTOR Pathway: Revised the text and replaced the figure with a more informative depiction.
4.5. The Keap1-Nrf2 Pathway: Refined the text and enhanced the figure for greater visual impact.
4.6. The Wnt/β-catenin Pathway: Added more details to the text and enriched the figure with additional insights.
4.7. The VEGF/VEGFR Pathway: Adjusted figure font; the text remained unchanged.
4.8. The NF-κB Pathway: Significantly revised the text and introduced a more detailed figure.
4.9. The PD-1/PD-L1 Pathway: Expanded the text and included a new figure to address your request.
4.10. The Apoptosis Pathway: Updated the text and introduced a corresponding figure for clarity.
4.11. The HIF Pathway: Revised text details and replaced the figure for consistency.
4.12. The Notch Signaling Pathway: Enhanced both the textual explanation and the figure.
4.Proofreading for Typos:
A comprehensive proofreading of the manuscript was conducted, and any identified typographical errors were corrected to improve the overall readability and precision of the document.
We trust these revisions address your concerns thoroughly and enhance the manuscript's scientific rigor and presentation. Thank you once again for your valuable suggestions, which have significantly contributed to strengthening the quality of our work.

Reviewer 3 Report
Comments and Suggestions for Authors
The authors of the review article entitled Genetic Blueprints in Lung Cancer: Foundations for Targeted 2 Therapies written a comprehensive review with updates in this field.
Minor recommendations before publication
1. If the authors can write about the past types frequencies and the actual one taking into account the smoking chances (cigarettes without filter, with filter and now electronically)
2. I recommend to organized in a separate Table, after the pathways described, the actionable genes - were target therapies are available
3. I recommend to write also about the promising technique - optical genome mapping
Author Response
We sincerely appreciate your thoughtful feedback and the opportunity to enhance our manuscript based on your insightful suggestions. Below, we provide a detailed response to your recommendations and outline the corresponding revisions made to the manuscript.
- If the authors can write about the past types frequencies and the actual one taking into account the smoking chances (cigarettes without filter, with filter and now electronically)
Thank you for highlighting the importance of discussing smoking trends and their impact on lung cancer. In response, we have expanded the manuscript to address historical and current smoking practices, including the evolution from non-filter to filter cigarettes and the emergence of electronic cigarettes. This discussion encompasses the distinct risks associated with each, including their differential impact on lung cancer subtypes and the long-term uncertainties surrounding e-cigarettes and vaping. The relevant information can be found in lines 79–90, page 3.
- I recommend to organized in a separate Table, after the pathways described, the actionable genes - were target therapies are available :
While we did not create a separate table for actionable genes, we integrated more detailed discussions of target therapies within the sections on molecular pathways. This approach allows for a more contextual understanding of how actionable genes align with specific signaling pathways. Additionally, we updated the text and figures for all molecular pathways to enhance clarity and provide a comprehensive depiction of their mechanisms and associated therapeutic options. Below is a summary of the specific revisions made to the molecular pathway sections:
4.1.The MAPK Pathway: Revised text and replaced the figure with an updated version.
4.2. The JAK-STAT Pathway: Expanded textual details and enhanced figure clarity with additional elements.
4.3. The LKB1/AMPK/mTOR Pathway: Updated both the text and figure for improved logical flow and detail.
4.4. The EGFR/PI3K/Akt/mTOR Pathway: Revised the text and replaced the figure with a more informative depiction.
4.5. The Keap1-Nrf2 Pathway: Refined the text and enhanced the figure for greater visual impact.
4.6. The Wnt/β-catenin Pathway: Added more details to the text and enriched the figure with additional insights.
4.7. The VEGF/VEGFR Pathway: Adjusted figure font; the text remained unchanged.
4.8. The NF-κB Pathway: Significantly revised the text and introduced a more detailed figure.
4.9. The PD-1/PD-L1 Pathway: Expanded the text and included a new figure to address your request.
4.10. The Apoptosis Pathway: Updated the text and introduced a corresponding figure for clarity.
4.11. The HIF Pathway: Revised text details and replaced the figure for consistency.
4.12. The Notch Signaling Pathway: Enhanced both the textual explanation and the figure.
- I recommend to write also about the promising technique - optical genome mapping
We appreciate your suggestion to include information about optical genome mapping as a promising technique. Accordingly, we have added a section that discusses the advantages of OGM over NGS methods, emphasizing its ability to detect structural variants that are often missed by short-read sequencing. We highlight how OGM complements existing methods by analyzing ultra-high molecular weight DNA and providing a high-resolution view of SVs, including those crucial to understanding intratumoral heterogeneity and metastatic progression in NSCLC. The relevant content is located in lines 688–710, page 23.
We hope these revisions address your concerns comprehensively and enhance the manuscript's scientific rigor and clarity. Thank you once again for your valuable feedback, which has significantly contributed to the quality and depth of our work.

Round 2
Reviewer 1 Report
Comments and Suggestions for Authors
Suggested to Accept in present form